# Characterization of the Protective Immune Responses Conferred by Recombinant BCG Overexpressing Components of *Mycobacterium tuberculosis* Sec Protein Export System

**DOI:** 10.3390/vaccines10060945

**Published:** 2022-06-14

**Authors:** Annuurun Nisa, Claudio Counoupas, Rachel Pinto, Warwick J. Britton, James A. Triccas

**Affiliations:** 1School of Medical Sciences, Faculty of Medicine and Health, The University of Sydney, Camperdown, NSW 2006, Australia; a.nisa@rutgers.edu (A.N.); c.counoupas@centenary.org.au (C.C.); rachel.pinto@sydney.edu.au (R.P.); 2Tuberculosis Research Program at the Centenary Institute, The University of Sydney, Sydney, NSW 2006, Australia; w.britton@centenary.org.au; 3Sydney Institute for Infectious Diseases and the Charles Perkins Centre, The University of Sydney, Camperdown, NSW 2006, Australia; 4Department of Clinical Immunology, Royal Prince Alfred Hospital, Camperdown, NSW 2005, Australia

**Keywords:** tuberculosis, vaccines, secretion system, recombinant BCG

## Abstract

*Mycobacterium bovis* Bacillus Calmette-Guérin (BCG) is the only approved vaccine against tuberculosis (TB). However, its efficacy in preventing pulmonary TB in adults is limited. Despite its variable efficacy, BCG offers a number of unique and beneficial characteristics, which make it suitable as a vaccine vehicle to express recombinant molecules. In *Mycobacterium tuberculosis*, the general Sec pathway is an essential cellular process, and it is responsible for exporting the majority of proteins across the cytoplasmic membrane, including potent immune-protective antigens, such as members of the antigen 85 (Ag85) complex. We engineered BCG to overexpress the *M. tuberculosis* SecDFG proteins in order to improve the efficiency of the Sec-dependent export system and, thus, enhance the secretion of immunogenic proteins. BCG^SecDFG^ displayed increased intracellular survival within macrophages in vitro and greater persistence in the lymphoid organs of vaccinated mice than parental BCG. In addition, vaccination with BCG^SecDFG^ generated higher numbers of IFN-γ-secreting T cells in response to secreted mycobacterial antigens compared to BCG, particularly members of the Ag85 complex. Furthermore, vaccination with BCG^SecDFG^ significantly reduced the bacterial load in the lungs and spleens of *M. tuberculosis*-infected mice, which was comparable to the protection afforded by parental BCG. Therefore, the modification of protein secretion in BCG can improve antigen-specific immunogenicity.

## 1. Introduction

*Mycobacterium tuberculosis*, the causative agent of TB, continues to spread due to the delay in detecting patients with active pulmonary TB [1] and the existence of a large reservoir of latently infected individuals, whose infection can reactivate over their lifetime. The current TB vaccine BCG is protective against severe forms of TB in children (meningitis and miliary TB), but confers inconsistent and variable protection against pulmonary TB in adolescents and adults, which is the transmissible form of the disease [2,3]. Previous sensitization with environmental mycobacteria or *M. tuberculosis* infection is considered one of the main factors associated with BCG lower efficacy against pulmonary tuberculosis [4]. This failure of BCG to provide consistent protection against adult pulmonary TB makes new and more effective TB vaccines critically important. Although BCG’s protective efficacy is variable in humans, it remains the sole approved vaccine against TB. The capacity of BCG to provide protective immunity against TB in children, as well as its potent immunostimulatory capacity [5], has maintained the interest in developing BCG into a novel vaccine vehicle with the capacity to express recombinant antigens and other immunostimulatory molecules to provide a more effective TB vaccine [5].

Similar to many other bacterial pathogens, *M. tuberculosis* is widely known for its capacity to produce and secrete numerous virulent proteins into the extracellular environment [6,7]. All *Mycobacteria* sp. possess two conserved protein export pathways, including the general secretion (Sec) and the twin-arginine translocation (Tat) pathways [8]. The pathogenic *M. tuberculosis* has two additional specialized protein-export systems: the accessory SecA2 system [9] and the type VII secretion system (T7SS), which encodes five different types of specialized early secretory system (ESX-1 to ESX-5) [10]. The Sec export pathway is a post-translational process system present in all bacteria; it is a primary route for exporting proteins to the cytoplasmic membrane and beyond [8,11]. Many mycobacterial proteins with roles in bacterial virulence are exported by the Sec system [12], including LspA [13], LpqH (19 kDa protein) [14], LppX [15], and LprG [16]. A comprehensive analysis revealed that the majority of the exported proteins (62%) identified in the culture filtrate of *M. tuberculosis* H37Rv were exported through the Sec pathway [17]. The major proteins exported by this pathway include members of the antigen 85 (Ag85) complex, the most abundant secreted *M. tuberculosis* proteins, which are potent immunoprotective antigens [18,19]. The Ag85 complex comprises three homologous major secretory proteins of *M. tuberculosis*, which have been the focus of extensive research: Ag85A (Rv3804c), Ag85B (Rv1886c), and Ag85C (Rv0129c) [20,21]. Because of their ability to induce a strong T-cell proliferation response [22], a large number of studies have utilized the Ag85 complex to develop vaccines against TB infection in various forms, including recombinant BCG (rBCG), subunit protein, and DNA vaccines [23,24].

The Sec pathway in *M. tuberculosis* consists of a total of 16 protein components [12]. SecD and SecF, along with YajC, are accessory Sec proteins, which contribute to the efficiency of mycobacterial protein export [25]. SecG forms a highly conserved heterotrimeric protein complex with SecY and SecE, which creates a channel in the cytoplasmic membrane, through which cytoplasm-synthesized proteins are transported to the extracellular environment. Unlike SecY and SecE, SecG is not required for bacterial viability, but its presence increases translocation efficiency by stabilizing the SecYE complex or assisting the conformational changes in SecA, a central component of the Sec transport system [8]. In *E. coli*, the overexpression of SecDF led to the increased membrane association of SecA, which resulted in increased protein translocation efficiency [26].

We hypothesized that the overexpression of the SecD, SecF, and SecG proteins in BCG would improve the efficiency of Sec-dependent export and increase the secretion of immunogenic proteins into the extracellular environment. This would lead to the improved immune recognition of secreted antigens, which may, in turn, improve the protective effect of the vaccine. In this study, the in vitro and in vivo growth of the BCG-overexpressing auxiliary components of the *M. tuberculosis* Sec pathway (BCG^SecDFG^) was examined, as well as the induction of T-cell immunity following vaccination in mice. The protective efficacy of BCG^SecDFG^ against aerosol *M. tuberculosis* challenge in mice was also determined.

## 2. Materials and Methods

### 2.1. Bacterial Strains and Culture Conditions

*M. bovis* BCG Pasteur (ATCC35734), *M. tuberculosis* H37Rv (ATCC27294; BEI Resources, NIAID, NIH, NR-13648), and the recombinant BCG strains were cultured in Middlebrook 7H9 (Difco™ BD, Franklin Lakes, NJ, USA) broth supplemented with albumin-dextrose-catalase (ADC; 10% *v/v*), Tween-80 (0.05% *v/v*) and glycerol (0.2% *v/v*) at 37 °C. To enumerate the bacterial numbers, cultures were plated onto Middlebrook 7H11 (Difco™ BD) agar with oleic-acid-albumin-dextrose catalase (OADC; 10% *v/v*) and glycerol (0.5% *v/v*). Antibiotic kanamycin (Sigma-Aldrich, St. Louis, MO, USA) was added to media when required at final concentration of 50 μg/mL.

### 2.2. Generation of BCG^SecDFG^ Strains

The construction of recombinant BCG overexpressing the SecDFG fusion protein was performed using the pMOD12-expression plasmid, which contains two independent and constitutively active promoters from BCG, HSP60, and HSP70, with two disparate multiple-cloning sites (MCSs) [27]. The DNA fragment codings for *M. tuberculosis* his-tagged SecD (Rv2587c) and SecF (Rv2586c) were inserted into the MCS1 of pMOD12 under the control of Phsp60, whereas c-Myc-SecG encoding-gene (Rv1440) was ligated into the MCS2 of the plasmid under Phsp70 promoter. The resultant plasmid was used to transform competent BCG and recombinants selected after plating on solid 7H11 media with kanamycin (50 μg/mL). The recombinant strain generated was called BCG^SecDFG^.

### 2.3. Western Blot

BCG cell lysates were prepared from exponential phase cultures using a Mini BeadBeater (BioSpec Products, Bartlesville, OK, USA). Cell lysates and concentrated culture supernatants (20 μg) were separated on a two phase 4%/12% or 4%/15% SDS-PAGE stacking/resolving gel electroblotted onto a nitrocellulose PVDF membrane (GE Healthcare, Chicago, IL, USA) and his-tagged SecDF fusion protein was detected with nickel-conjugated horseradish peroxidase (Ni-HRP His detect; SeraCare, Milford, MA, USA). The c-Myc tagged SecG or Ag85B proteins were detected using rabbit monoclonal anti-c-Myc (Abcam, Cambridge, UK) and rabbit polyclonal anti-Ag85B antibodies, respectively. Blots were developed with SuperSignal West Pico Chemiluminescent substrate (Thermo Fisher Scientific, Waltham, MA, USA) and visualized using a ChemiDoc MP imaging system (Bio-Rad, Hercules, CA, USA).

### 2.4. Macrophage Infection

RAW 264.7 murine macrophage cells (ATCC TIB-71) were plated at 2 × 10^5^ cells/well and BCG^pMOD12^ or BCG^SecDFG^ were then added to the cells at a multiplicity of infection (MOI) of 1:1 or 1:5 in RPMI media supplemented with 10% fetal bovine serum (FBS). Cell culture was incubated (37 °C, 5% CO_2_) for 4 h, extracellular bacteria removed by extensive washing, and colony forming units (CFU) determined by lysis of cells and growth on supplemented 7H11 agar medium.

### 2.5. Vaccination of Mice and M. tuberculosis Challenge

All murine experiments were performed in strict accordance with the approval granted by the University of Sydney Animal Ethics Committee (K75/9-2012/3/5846). Female C57BL/6 mice, aged 6–8 weeks old, were purchased from the Animal Resources Center (Perth, Australia). For the adoptive transfer studies, P25-TCR-transgenic mice with transgenic expression of a TCR specific for I-A^b^-Ag85B_240-254_ peptide generated on a C57BL/6 were utilized. For subcutaneous (s.c.) vaccine administration, mice were anaesthetized with gaseous isofluorane (4%) and injected at the base of tail with either 5 × 10^5^ CFU of BCG^SecDFG^, the same amount of BCG carrying the empty plasmid vector (BCG^pMOD12^), or left unvaccinated as negative control. Eight weeks after vaccination, mice were challenged with *M. tuberculosis* H37Rv in an inhalation exposure system using a Middlebrook airborne infection apparatus (Glas-Col, Terre Haute, IN, USA) with an infective dose of 100 CFU. Four weeks after challenge, serial dilutions of lungs and spleen homogenates were plated on supplemented Middlebrook 711 agar (Difco™ BD) to enumerate the bacterial loads. All experiments involving *M. tuberculosis* were performed in a Physical Containment Level 3 laboratory at the Centenary Institute.

### 2.6. Organ Collection and Processing

For isolation of lung leukocytes, lung tissue in complete RPMI media (L-glutamine and 25 mM Hepes (Invitrogen, Carlsbad, CA, USA), FCS (10% *v/v*), 2-mercaptoethanol (50 μM; Sigma-Aldrich) and PenStrep (100 U/mL; Invitrogen)) was digested with collagenase IV (50 U/mL; Sigma-Aldrich, St. Louis, MO, USA) and DNAse I (13 μg/mL; Sigma-Aldrich) at 37 °C for 45 min prior to homogenization (using a GentleMACS Dissociator, Miltenyi Biotec, Bergisch Gladbach, Germany) and multiple filtration steps. Lymph nodes and spleens were homogenized by passing through a 40-micrometer cell strainer (BD Biosciences, San Jose, CA, USA) in RPMI media and then pelleted by centrifugation (5 min, 4 °C, 1500 rpm). Erythrocytes were removed using ACK lysis buffer and single-cell suspensions were prepared and counted by Trypan Blue (0.04%) exclusion, followed by dilution in desired concentration. For adoptive transfer studies, splenocytes from p25-TgTCR (p25) mice were prepared and labeled with CFSE (Molecular Probes-Invitrogen, Eugene, OR, USA), as previously described [28]. C57BL/6 mice (CD45.2) received intravenously (i.v.) 5 × 10^5^ CFSE-labeled p25 splenocytes (CD45.1) and, the following day, they were immunized with 5 × 10^5^ CFU of BCG^pMOD12^, BCG^SecDFG^, or PBS. Five days after vaccination, spleen and lymph nodes were collected to prepare single cell suspensions.

### 2.7. IFNγ ELISPOT

ELISPOT filter plates (Millipore, Darmstadt, Germany) were coated (overnight, 4 °C) with anti-IFNγ antibody (10 μg/mL) in PBS, washed, and blocked with complete RPMI media for 2 h at 37 °C. Cell suspensions were plated at 2 × 10^5^ cells per well and P25 peptide, concentrated BCG^SecDFG^ supernatants, or Ag85A, Ag85B, Ag85C, CFP, and MPT32 antigens (10 μg/mL; BEI Resources (Manassas, VA, USA) were added. Plates were incubated for 20 h, at 37 °C in 5% CO2. Plates were washed with PBS and IFN-γ spots were detected, as previously described [29]. Developed spots were counted using an automated ELISPOT reader (AID EliSpot Reader software v 6.0; AID GmbH, Strassberg, Germany) and reported as frequency per million cells.

### 2.8. Flow Cytometry

For the analysis of antigen-specific cytokine production by T cells, single-cell suspensions (5 × 10^6^ cells/mL) were stimulated for 3–4 h (37 °C, 5% CO_2_) with P25 peptide, CFP, or concentrated BCG^SecDFG^ supernatant (10 μg/mL) in the presence of Brefeldin A (10 μg/mL; Sigma-Aldrich). Cell suspensions were incubated with CD16/CD32 (2.4G2; BD Biosciences) to block Fc receptors and then stained with Fixable Blue Dead Cell stain (Thermo Fisher Scientific) and fluorophore-conjugated monoclonal antibodies against surface markers (Appendix A). Cells were fixed and permeabilized using BD Cytofix/Cytoperm kit (BD Biosciences). Fluorophore-conjugated monoclonal antibodies against intracellular markers (Appendix A) were diluted in BD Perm/Wash (BD Biosciences) for intracellular staining. Immunostained cells or beads were fixed in 10% buffered formalin prior to data acquisition using an LSR-Fortessa analyzer (BD Biosciences). Acquired data was analyzed using FlowJo analysis software (Treestar Inc., Ashland, OR, USA) using the gating strategy illustrated in Appendix A. A Boolean gate combination was used to calculate the frequency of single- or multiple-cytokine-positive cell subsets.

### 2.9. Statistical Analysis

Statistical analysis was performed using GraphPad Prism 7 software (GraphPad Software, La Jolla, CA, USA). The significance of differences between experimental groups was evaluated by one- or two-way analysis of variance (ANOVA), with pairwise comparison of multi-grouped data sets achieved using Tukey’s honest significant difference (HSD) post hoc test. The results were considered significant when the *p* values were ≤0.05.

## 3. Results

### 3.1. Construction and Characterization of rBCG Strains Overexpressing the SecDFG Components from the M. tuberculosis Sec Export System

A dual-promoter expression plasmid, pMOD12, was used to construct the rBCG strain overexpressing the recombinant proteins SecD, SecF, and SecG [27], termed BCG^SecDFG^ (Figure 1A). Bands of the expected size for SecD (60.23 kDa) and Sec F (47.01 kDa) were detected by immunoblot compared to parental BCG, confirming the overexpression in the recombinant strains (Figure 1B). However, the expression of SecG (8.16 kDa) was not detected in the isolates tested (Figure 1B), possibly due to the limitations of detecting low-molecular-weight products [30]. To confirm the functional impact of the SecDFG overexpression, an immunoblot was performed to determine the presence of the immunodominant Ag85 proteins in the extracellular milieu. A band with a size of ~34 kDa was detected with higher intensity in the BCG^SecDFG^ clones compared to normal BCG (Figure 1C), suggesting that the overexpression of the SecDFG altered the secretion of Ag85 proteins to the extracellular environment. The BCG^SecDFG^ displayed a reduced growth rate in the culture compared to the wild-type BCG, suggesting the impact of protein overexpression on growth (Figure 1D).

### 3.2. Increased In Vitro Intracellular Persistence and In Vivo Immunity by BCG^SecDFG^

We hypothesized that the improved secretion of proteins may alter the interaction with host cells. This could be due to increased secretion of virulence/persistence determinants or enhanced cell-association via improved export of receptor-binding molecules, such as Ag85B [31]. RAW264.7 murine macrophage cells were infected with BCG^pMOD12^ or BCG^SecDFG^ and the bacterial load was determined at various timepoints. Although there was no difference in the initial uptake by the BCG^SecDFG^ s at 4 h post-infection compared to the cells infected with the parental BCG, the BCG^SecDFG^ displayed a significant increase in recovered CFU at 168 h following infection at an MOI of 1 (Figure 1E) and, at 72 h, at an MOI of 5 (Figure 1F) compared to the BCG^pMOD12^. Thus, the SecDFG expression appeared to alter the capacity of the BCG to survive/replicate long-term within the host cells.

To determine the impact of the SecDFG on the in vivo persistence, the mice were vaccinated with either the BCG^pMOD12^ or the BCG^SecDFG^, and the kinetics of the BCG growth were measured by enumerating the bacterial load in the organs. In the draining lymph nodes (DLNs), similar numbers of CFU were observed in both vaccinated groups at day 1 following the vaccination. However, the BCG^SecDFG^ demonstrated greater persistence than the parental strain in the DLNs and spleens at day 7 post-vaccination (Figure 2A). In the lungs and spleens, the numbers of the BCG^SecDFG^ declined to undetectable levels by day 84, but remained elevated in the DLNs, indicating enhanced dissemination to this site.

To determine whether the difference in the expressions of the secretion-system components in the BCG^SecDFG^ had an effect on the immune response generated, a panel of secreted mycobacterial proteins (Ag85A, Ag85B, Ag85C, and MPT32) was used to stimulate the lymphocytes from the BCG^pMOD12^ and the BCG^SecDFG^-vaccinated mice, and the frequency of the antigen-specific IFNγ-producing cells was determined by ELISPOT. In the spleens, the increased immunogenicity induced by the BCG^SecDFG^ compared to the parental BCG was apparent, particularly for the three Ag85 antigens, at day 28 post-vaccination (Figure 2B). The IFN-γ responses to the CFP were highest in the BCG^SecDFG^-vaccinated mice at 28 days, demonstrating the persistent immune responses to these antigens after the vaccination of the mice. In the DLNs, the BCG^SecDFG^ responses were typically compared to those of the control mice, but these differences only reached significance after the stimulation with the *M. tuberculosis* CFP (Appendix A).

The increased persistence of the BCG^SecDFG^ in the lymphoid organs, together with the increased IFN-γ responses in the vaccinated mice, suggested the vaccine may induce the increased priming of T cells. To test this, p25 transgenic CD4^+^ T cells specific for the dominant P25 epitope of Ag85B (CD45.1^+^) were transferred to recipient C57BL/6 mice (CD45.2^+^), and the next day, the mice were vaccinated s.c. with the BCG strains (Figure 3A). The mice that received the BCG^SecDFG^ had significantly higher numbers of p25 CD4^+^ T cells in the LNs and spleens compared to the mice that received the BCG^pMOD12^ (Figure 3B). A similar pattern of cytokine-producing P25 CD4^+^ T cells was observed from the LN cells, with both BCG strains generating a high proportion of triple-positive IFN-γ^+^IL-2^+^TNF^+^ CD45.1^+^ T cells at day 6 post-vaccination (Figure 3C).

### 3.3. Protective Efficacy of BCG^SecDFG^ against Aerosol M. tuberculosis Infection

Next, we determined whether the modification of the protein secretion by the BCG affected the protective efficacy. Ten weeks following s.c. immunization with BCG^pMOD12^ or BCG^SecDFG^, the mice were challenged with a low dose of *M. tuberculosis* H37Rv and, 4 weeks later, the bacterial numbers in the lungs (Figure 4A) and spleens were enumerated (Figure 4B). In the lungs, both the BCG^pMOD12^ and the BCG^SecDFG^ vaccinations provided significant protection compared to the unvaccinated mice at 4 weeks post-*M. tuberculosis* challenge (Figure 4A). There was no significant difference between the BCG^SecDFG^ and the parental strain in the capacity to reduce the *M. tuberculosis* burden, with a similar reduction in *M. tuberculosis* in the lungs (1.45 and 1.31 log_10_, respectively). In the spleens, the BCG^SecDFG^ provided increased protection, with a reduction in *M. tuberculosis* by approximately 0.20 log_10_ CFU compared to the control BCG, although this difference was not statistically significant (Figure 4B). However, the BCG^SecDFG^-vaccinated mice generated an increased percentage of IFN-γ^+^IL-2^+^TNF^+^ lung CD4^+^ T cells following the *M. tuberculosis* aerosol challenge compared with the mice that received the parental BCG strain (Figure 4C). Overall, these results indicate that the BCG^SecDFG^ was protective against *M. tuberculosis* infection and maintained improved vaccine-specific immunity after the challenge.

## 4. Discussion

BCG is the only approved vaccine against TB, but in low and middle-income countries, the protection provided by BCG is insufficient to contain transmissions. Hence, there is an urgent need to improve BCG efficacy to make it more effective and long-lasting in controlling and preventing TB infection. In this report, we sought to improve BCG immunity by altering its protein-secretion system, thus potentially increasing the quality and magnitude of the immune responses against secreted proteins [32]. Indeed, we observed an increased secretion of Ag85 proteins, which are key immunogenic proteins of mycobacteria [19]. Previous studies investigated the impact of the heterologous expression of proteins from the ESX-1 secretion machinery on the quality and magnitude of the immune responses against *M. tuberculosis* infection [32,33]. A recombinant BCG-overexpressing ESX-1 was shown to provide superior protection against highly virulent *M. tuberculosis* relative to parental BCG, stimulating increased proportions of polyfunctional CD4^+^ T cells and strong CD8^+^ T-cell-effector responses [32,33]. These findings support the concept of altering the protein secretion systems in BCG to increase the transport of immunogenic proteins to enhance protective efficacy.

In vitro infection studies using a murine macrophage cell line revealed a greater survival for BCG^SecDFG^ over time (Figure 1). The observed increase in the bacterial numbers observed at 72 h post-infection may have been caused by the increased replication of the bacteria inside the host cells. Protein secretion plays a crucial role in modulating the interactions between bacteria by altering host-cell functions, including vesicular trafficking and host immune responses [34]. During intracellular growth, *M. tuberculosis* produces virulence factors that actively modify the phago-lysosome to create conditions conducive to bacterial proliferation [35]. The proteins exported by the Sec system include several important virulence factors [12]; thus, the overexpression of SecDFG could result in the secretion of proteins that enhance the survival of BCG inside host cells.

We observed the greater persistence of the BCG^SecDFG^ in the DLN and the increased recovery of the BCG^SecDFG^ from the spleens of mice 7 days post-immunization (Figure 2), which was consistent with the enhanced BCG^SecDFG^ persistence observed in in vitro macrophage cell culture. It is known that following s.c. vaccination, the initial BCG replication occurs in the DLN, which results in higher bacterial dissemination to the spleen than to the lungs [36]. It has been proposed that BCGs chronically persist in the draining lymph nodes of mice after s.c. vaccination [37], and may function as a reservoir for dissemination to the spleen and lungs [38]. Some studies have reported that BCG disseminates to the lungs in low numbers around 20 weeks following s.c. vaccination [36]. However, in our study, we were not able to detect BCG or BCG^SecDFG^ in the lungs at any timepoints. Overall, these data suggest that the overexpression of *M. tuberculosis* SecDFG protein in BCG improved the bacterial persistence in the murine organs, particularly in the DLNs, which correlates with the greater dissemination to the spleen.

The greatest impact on the immune responses induced by the BCG^SecDFG^ vaccination was observed following stimulation with members of the Ag85 complex (Figure 2). This was also more evident upon recall with *M. tuberculosis* CFP. CFP contains many different proteins, resulting in various antigen exposures to immune cells, the most predominant of which are members of the Ag85 complex [39]. This also correlates with previous work demonstrating the enhanced immunogenicity of BCG-overexpressing Ag85B [40] and the role played by Ag85B in facilitating host-cell interactions [31]. The enhanced Ag85B secretion was further validated when examining the priming of the Ag85B-specific (p25) transgenic T cells, which demonstrated the increased proliferation of antigen-specific T cells (Figure 3), possibly reflecting a greater secretion of Ag85B in the BCG^SecDFG^-vaccinated mice. The analysis of the cytokines secreted by the p25 T cells showed that the BCG^SecDFG^ induced a higher proportion of polyfunctional CD4^+^ T cells to produce multiple cytokines, including IFN-γ, IL-2, and TNF (Figure 3). We also observed an increase in the frequency of multifunctional Th1-like immunity induced by BCG^SecDFG^ vaccination post-*M. tuberculosis*-challenge (Figure 4). Th1-type responses are recognized as essential for protection against mycobacterial diseases, including TB [41], and polyfunctional CD4^+^ T cells secreting IFN-γ, IL-2, and TNF have been proposed as markers of protective immunity [38]. Furthermore, the IL-2^+^TNF^+^ CD4^+^ T-cell population is described as having an effector-memory phenotype with high proliferative capacity [42,43]. Multiple studies have shown that IFNγ and TNF are required for TB immunity, although, by contrast, several other studies have also shown that the magnitude of IFNγ-producing T cells alone does not correlate with the degree of protection against TB [44].

In accordance with the observed immunogenicity induced by the BCG^SecDFG^, the vaccine reduced the bacterial burden on both the lungs and the spleens of the *M. tuberculosis*-infected mice at 4 weeks post-challenge (Figure 4). Although the parental strain of the BCG also showed a decreased bacterial burden, the BCG^SecDFG^ displayed a trend toward greater protection in both organs. Previously, we showed that the overexpression of Ag85B in BCG does not improve the protective effect of BCG in mice, despite a marked increase in antigen-specific immunity [45]. Thus, it would be of interest to determine the efficacy of BCG^SecDFG^ in other pre-clinical models, since the improved secretion of mycobacterial proteins appears to have an impact on BCG’s protective efficacy [46]. Overall, this study demonstrates that the modification of the protein-secretion apparatus in BCG can serve to enhance protein secretion and influence bacterial persistence and immunogenicity. The assessment of this vaccine in other pre-clinical models, as well as the combination of this strategy with other modifications to BCG, may serve to generate a suitable candidate to improve the efficacy of BCG vaccination in humans.

## Figures and Tables

**Figure 1 vaccines-10-00945-f001:**
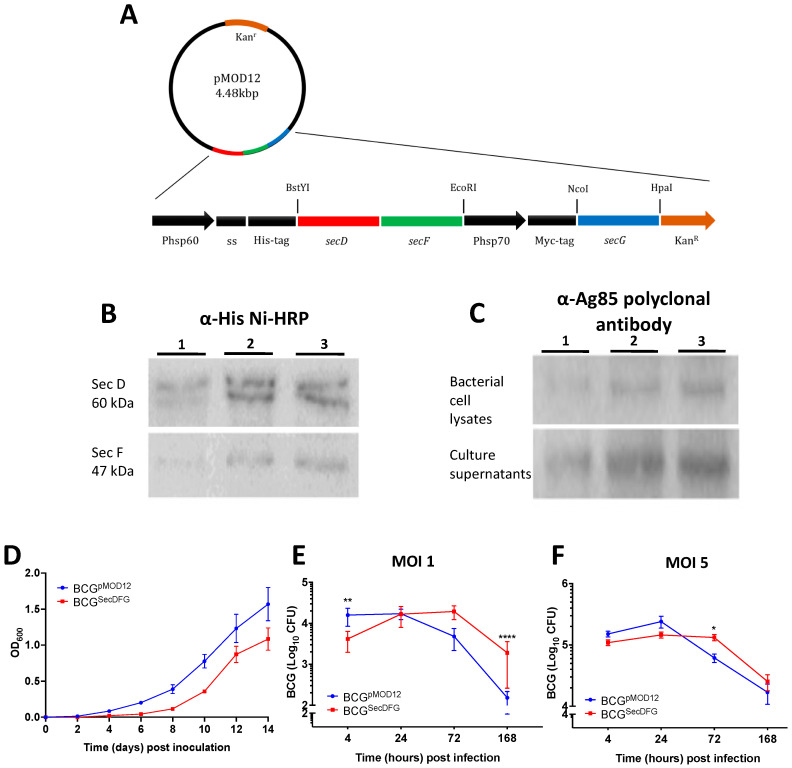
Characterization of recombinant BCG strains overexpressing *M. tuberculosis* SecDFG protein. (**A**). A schematic diagram of the recombinant prokaryotic expression plasmid, pMOD12-SecDFG. (**B**). Cell lysates (20 μg) were prepared, separated on a 12% SDS-PAGE gel, and transferred to nitrocellulose membrane, and polyhistidine-tagged SecDF was detected using Ni-HRP conjugate for secondary detection of his-tagged proteins (Lane 1: BCG^pMOD12^, Lane 2: BCG^SecDFG^ clone 1, Lane 3: BCG^SecDFG^ clone 2). (**C**). Culture supernatants and cell lysates (20 μg) were examined by immunoblot using anti-Ag85 complex polyclonal antibody. (**D**). Bacterial growth in 7H9 medium (OD_600_). Values represent mean ± SD of two independent cultures. E-F. RAW 264.7 murine macrophage cells were infected with rBCG strains a MOI of 1 (**E**) or 5 (**F**) and CFU was determined at indicated timepoints. Data are the means ± SEM (*n* = 6) and are representative of two independent experiments. Statistical significance between groups was determined by ANOVA with Sidak’s multiple-comparisons test (* *p* < 0.05; ** *p* < 0.01; **** *p* < 0.0001).

**Figure 2 vaccines-10-00945-f002:**
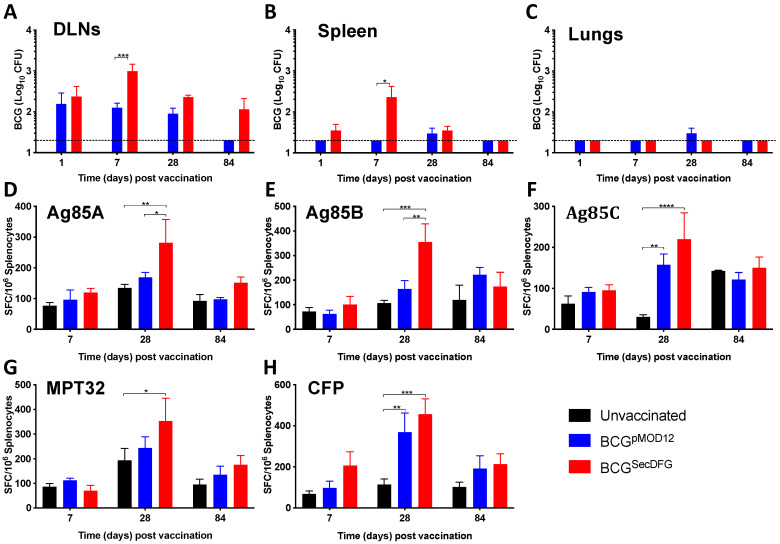
Persistence and immunogenicity of BCG^SecDFG^ in mice. C57BL/6 mice were left unvaccinated (black bars), or vaccinated s.c. with 5 × 10^5^ CFU BCG^pMOD12^ (blue bars) or BCG^SecDFG^ (red bars), and the bacterial load was determined in the DLNs (**A**), spleens (**B**), and lungs (**C**) at days 1, 7, 28, and 84. Splenocytes from days 7, 28, and 84 were stimulated ex vivo with 10 μg/mL of Ag85A (**D**), Ag85B (**E**), Ag85C (**F**), MPT32 (**G**), or CFP (**H**) for approximately 20 h at 37 °C. The number of antigen-specific IFN-γ-secreting cells was enumerated by ELISPOT. The data are the means ± SEM and representative of two independent experiments. Detection limit of the assay shown by the dotted line (20 CFU/organ). Statistical significance was determined by ANOVA with Tukey’s multiple-comparisons test (* *p* < 0.05; ** *p* < 0.01; *** *p* < 0.001; **** *p* < 0.0001).

**Figure 3 vaccines-10-00945-f003:**
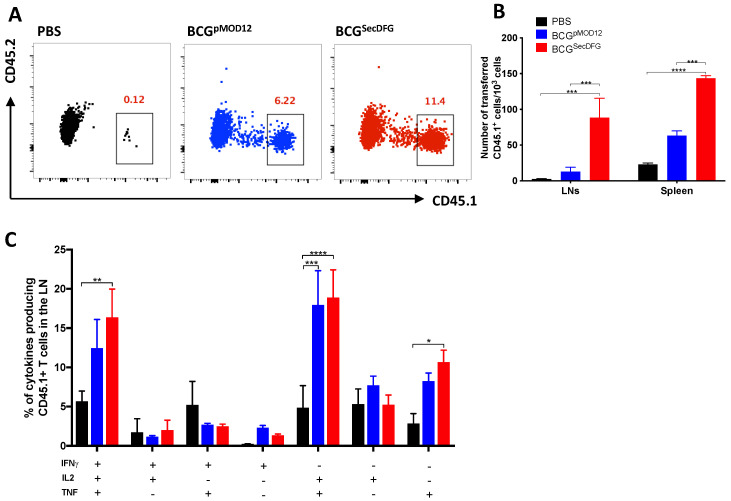
Expansion of transferred p25-specific CD4^+^ T cells following BCG^SecDFG^ vaccination. Approximately 5 × 10^5^ CFSE-labeled p25 splenocytes (CD45.1^+^) were transferred via i.v. injection to C57BL/6 (CD45.2^+^) recipient mice (*n* = 4) prior to vaccination. The following day, mice were vaccinated s.c. with 5 × 10^5^ CFU of BCG^pMOD12^ or BCG^SecDFG^, or left unvaccinated. Six days after the vaccination, the numbers of lymphocytes were enumerated in the DLNs and spleens. Representative flow plots are shown (**A**), together with CD45.1^+^ P25-specific CD4^+^ T numbers (**B**), as measured by flow cytometry. (**C**). Intracellular cytokine staining was performed on lung cells at 4 weeks post-infection after re-stimulation with CFP (10 μg/mL) and Brefeldin A, and the frequency of IFN-γ, IL-2, and/or TNF positive CD4^+^ T cells was assessed. Data are representative of two independent experiments. Statistical significance was determined by ANOVA with Tukey’s multiple-comparisons test (* *p* < 0.05; ** *p* < 0.01; *** *p* < 0.001; **** *p* < 0.0001; gating strategy is shown in Appendix A.

**Figure 4 vaccines-10-00945-f004:**
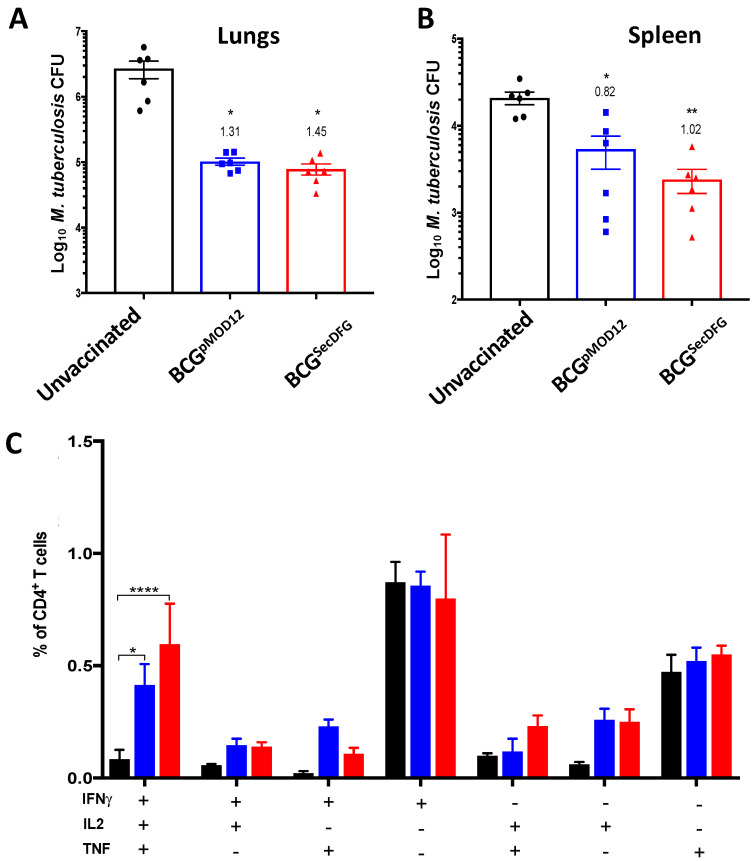
Protection afforded by BCG^SecDFG^ vaccination against aerosol *M. tuberculosis* infection in C57BL/6 mice. C57BL/6 mice (*n* = 6) were immunized s.c. with 5 × 10^5^ CFU of BCG^pMOD12^ (blue) or BCG^SecDFG^ (red), or PBS as control (black), and 10 weeks post-immunization, mice were challenged with ~100 CFU of *M. tuberculosis* H37Rv via aerosol route. The bacterial loads in lungs (**A**) and spleens (**B**) were enumerated following culture on Middlebrook 7H11 agar. (**C**). Intracellular cytokine staining was performed on lung cells at 4 weeks post-infection after re-stimulation with CFP (10 μg/mL) and Brefeldin A, and the frequency of IFN-γ, IL-2, and/or TNF-positive CD4^+^ T cells was assessed by flow cytometry. The data are the means ± SEM and representative of two independent experiments. The numbers on top of the bar graphs indicate the Log protection compared to the unvaccinated group. Statistical significance was determined by ANOVA with Tukey’s multiple-comparisons test (* *p* < 0.05; ** *p* < 0.01; **** *p* < 0.0001).

## Data Availability

Not applicable.

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
