# Peer review of "Characterization of the Protective Immune Responses Conferred by Recombinant BCG Overexpressing Components of Mycobacterium tuberculosis Sec Protein Export System"

_vaccines, 2022, doi:10.3390/vaccines10060945_

Round 1

Reviewer 1 Report

The manuscript “Characterization of the protective immune responses conferred  by recombinant BCG overexpressing components of Mycobac terium tuberculosis Sec protein export system” describes the costruction BCG to overexpress the M. tuberculosis SecDFG proteins in order to improve the efficiency of Sec-dependent  export system and thus enhancing the secretion of immunogenic proteins. The results are quite intersting and importante to improve the BCG vaccine for TB. Thus I have minor revison before accept the manuscript for publication.

1. In the item  Vaccination of mice and M. tuberculosis challenge of metodology, the authors could explain better the groups, for example: G1-BCGpMOD12,; G2- BCGSecDFG, G3-Unvaccinated

Author Response

Point 1. In the item  Vaccination of mice and M. tuberculosis challenge of metodology, the authors could explain better the groups, for example: G1-BCGpMOD12,; G2- BCGSecDFG, G3-Unvaccinated

Response 1. An updated description of the groups is now included on lines 109-110 and 138-139.

We thank the reviewer for the positive overall feedback.

Reviewer 2 Report

I have reviewed the paper by Nisa et al.

The paper is elegant and well written.

The following, need to be fixed though.

In the introduction, the limitation for BCG as a vaccine has to be clarified to be in regards to protection against Pulmonary TB.

The major issue with this paper is the utilization of murine macrophages, when they could have done the experiment using a human cell line. Was this done in order to be able to make some correlation with the in vivo experiments?

The recombinant strain has more growth in vitro and in vivo, offering higher protection in Lungs. These are excellent findings

Author Response

Point 1. In the introduction, the limitation for BCG as a vaccine has to be clarified to be in regards to protection against Pulmonary TB.

Response 1. The variable protection by BCG against pulmonary TB is now explained in lines 41-43.

Point 2. The major issue with this paper is the utilization of murine macrophages, when they could have done the experiment using a human cell line. Was this done in order to be able to make some correlation with the in vivo experiments?

Response 2. The reviewer is correct, we consider that it is more consistent to standardise the in vitro and in vivo models using murine system.

Point 3. The recombinant strain has more growth in vitro and in vivo, offering higher protection in Lungs. These are excellent findings

We thank the reviewer for the encouraging comment.

Reviewer 3 Report

This manuscript is a vaccine development study to enhance immunogenicity of BCG by releasing many secreted antigens out of TB through the Sec pathway. Antigen secretion was increased by overexpressing the Sec system in BCG. Although not very meaningful as a vaccine, it showed the possibility of enhancing immunogenicity through secreted antigen.

Line 109 ~ 118 It would be better to separate this paragraph into sections such as Immunoassay rather than "Generation of BCG strains".

Line 120-125 How can the authors distiguish between adhered and infected bacteria without antibiotic treatment?

Line 127 If the authors have performed the experiments it a BSL3 system, the authors should provide relevant information.

Combine Line 141(Organ collection) and Line 162 (Immunogenicity studies)

Fig 1B. Anti-his antibody was used, but the signal is also seen in #1 of Fig 1B. Although they are stronger in #2 and #3, the author need to explain it.

Line 213. What is Ni-HRP His-detect conjugate? Is it the seconday antibody or pull down assay?

Line 315. Please specify what the statistical comparison group is. For example, p<0.1 compared with unvaccinated group

Line 315 p<0.1 is statistic significant? You should explain your statistic analysis method and why 0.1 is statistically significant?

Author Response

Point 1. Line 109 ~ 118 It would be better to separate this paragraph into sections such as Immunoassay rather than "Generation of BCG strains".

Response 1. The paragraph is now its own section titled “Western Blot”.

Point 2. Line 120-125 How can the authors distinguish between adhered and infected bacteria without antibiotic treatment?

Response 2. We extensively wash cells and has previously shown that this effectively removes extracellular bacteria.

Point 3. Line 127 If the authors have performed the experiments it a BSL3 system, the authors should provide relevant information.

Response 3. The work in BSL3 is now stated on lines 142-144.

Point 4. Combine Line 141(Organ collection) and Line 162 (Immunogenicity studies)

Response 4. The organ processing section of the “immunogenicity study” paragraph has now been moved to the “organ collection section”.

Point 5. Fig 1B. Anti-his antibody was used, but the signal is also seen in #1 of Fig 1B. Although they are stronger in #2 and #3, the author need to explain it.

Response 5. As noted by the reviewer there is a clear signal in the rBCG groups, with a weak signal in the control group. This represents non-specific binding of the anti-His antibody, which is a known property of this reagent.

Point 6. Line 213. What is Ni-HRP His-detect conjugate? Is it the secondary antibody or pull-down assay?

Response 6. Ni-HRP was used as a secondary antibody. This is now better explained in the figure legend.

Point 7. Line 315. Please specify what the statistical comparison group is. For example, p<0.1 compared with unvaccinated group

Response 7. This is an error, the value should be p<0.05. This is now corrected.

Point 8. Line 315 p<0.1 is statistic significant? You should explain your statistic analysis method and why 0.1 is statistically significant?

Response 8. Apologies, the p<0.1 is a typographical error, all the one-star significance are p<0.05, this has now been fixed throughout the manuscript.

Reviewer 4 Report

In the current manuscript, Nisa et al. report the immune efficacy of recombinant BCG overexpressing the components of Mtb Sec proteins. The authors showed that the recombinant BCG had improved survival within the macrophages and in the lymphoid organs of vaccinated mice. The authors also showed that the vaccinated mice had a lower Mtb burden in the lungs and spleens. Overall, their conclusions are justified by their results; However, they should replace certain data with better figures and improve their figures. My minor issues:

1. Figure 1B and 1C are very poorly developed. The authors should either repeat the experiment or replace the current blots with better ones. 

2. Line: 201: Change 'imitations' to 'limitations'. 

3. Label the color lines with a legend in panel 1D-1F. 

4. Figure-4: In panels A and B, the comparisons are not clear and the p values on top of BCG vaccinated are confusing. Please check the stats and replace the p values. As of now, none of those values seems significant. 

Author Response

Point 1. Figure 1B and 1C are very poorly developed. The authors should either repeat the experiment or replace the current blots with better ones. 

Response 1. The difficulty in obtaining highly concentrated samples from rBCG strains, particularly supernatants,  together with the low quality of currently available reagent to detect his-tagged proteins using such samples, makes it difficult to generate very high quality blots. We have repeated this experiment a number of times with the same outcome, and consider this figure to be the most representative of our results.

Point 2. Line: 201: Change 'imitations' to 'limitations'. 

Response 2. This error has now been corrected

Point 3. Label the color lines with a legend in panel 1D-1F. 

Response 3. Labels are now added to panel 1D and 1E

Point 4. Figure-4: In panels A and B, the comparisons are not clear and the p values on top of BCG vaccinated are confusing. Please check the stats and replace the p values. As of now, none of those values seems significant. 

Response 4. The numbers on top of the groups represent the Log protection value. This is now stated in the figure legend.

Round 2

Reviewer 3 Report

No more major questions.